# Adversarial Machine Learning for NextG Covert Communications Using Multiple Antennas

**DOI:** 10.3390/e24081047

**Published:** 2022-07-29

**Authors:** Brian Kim, Yalin Sagduyu, Kemal Davaslioglu, Tugba Erpek, Sennur Ulukus

**Affiliations:** 1Department of Electrical and Computer Engineering, University of Maryland, College Park, MD 20742, USA; bkim628@umd.edu; 2Virginia Tech, National Security Institute, Arlington, VA 24061, USA; ysagduyu@vt.edu (Y.S.); terpek@vt.edu (T.E.); 3University Technical Services, Greenbelt, MD 20770, USA; kemal@ut-services.com

**Keywords:** deep learning, covert communications, signal classification, adversarial attack

## Abstract

This paper studies the privacy of wireless communications from an eavesdropper that employs a deep learning (DL) classifier to detect transmissions of interest. There exists one transmitter that transmits to its receiver in the presence of an eavesdropper. In the meantime, a cooperative jammer (CJ) with multiple antennas transmits carefully crafted adversarial perturbations over the air to fool the eavesdropper into classifying the received superposition of signals as noise. While generating the adversarial perturbation at the CJ, multiple antennas are utilized to improve the attack performance in terms of fooling the eavesdropper. Two main points are considered while exploiting the multiple antennas at the adversary, namely the power allocation among antennas and the utilization of channel diversity. To limit the impact on the bit error rate (BER) at the receiver, the CJ puts an upper bound on the strength of the perturbation signal. Performance results show that this adversarial perturbation causes the eavesdropper to misclassify the received signals as noise with a high probability while increasing the BER at the legitimate receiver only slightly. Furthermore, the adversarial perturbation is shown to become more effective when multiple antennas are utilized.

## 1. Introduction

Privacy is a fundamental problem in wireless communications due to the open and shared nature of wireless medium. An eavesdropper may overhear the communications intended between a transmitter and a receiver. The eavesdropper may pursue different objectives such as decoding transmissions or detecting whether there is an ongoing transmission, or not (e.g., for launching follow-up jamming attacks). The privacy of information regarding unapproved decoding has been extensively studied from both encryption-based security and information theory perspectives [1,2]. In this paper, we consider an eavesdropper that pursues the second objective, namely detecting an ongoing transmission for future adversarial purposes such as jamming to degrade the quality of communications.

*Covert communications* has been studied to hide information in noise where the main goal has been to reduce the signal-to-noise ratio (SNR) at the eavesdropper [3,4]. A fundamental bound has been demonstrated on the total transmit power over a given number of channel users while maintaining covert communications, generally known as the square-root law [5]; see also [6] for related work. In this paper, we study covert communications from an *adversarial machine learning* (AML) point of view. Overall, AML is an emerging field that studies machine learning (ML) in the presence of adversaries that may aim to manipulate the test and/or training pipelines of ML algorithms [7,8,9]. While the applications of AML have originated in the computer vision domain, there has been a growing interest in applying AML to wireless communications [10,11,12], including exploratory (inference) attacks [13,14], evasion (adversarial) attacks [15,16,17,18,19,20,21,22,23,24,25,26,27,28,29,30,31,32,33] and their extensions to secure and covert communications against eavesdroppers [34,35,36,37], causative (poisoning) attacks [38,39,40], membership inference attacks [41,42], Trojan attacks [43], and spoofing attacks [44,45,46,47].

We consider an eavesdropper with a deep learning (DL)-based classifier to detect an ongoing transmission where this classifier achieves a high accuracy for distinguishing the received signals from noise. We introduce a *cooperative jammer* (*CJ*) that has been extensively used in the physical layer security literature [48,49,50]. In this paper, the CJ transmits signals over the air at the same time as the transmitter with the purpose of fooling the eavesdropper’s classifier for covert communications. These signals from CJ corresponds to an evasion attack (or adversarial attack) in AML where evasion attacks have been used to manipulate wireless signal classification (in particular, modulation classification) [15,16,17,18,19,20,21,22,23,24,25,26,27,28], spectrum sensing [29], autoencoder communications [30], initial access [31], channel estimation [32], and power control [33]. In this paper, adversarial attack is used as a means of covert communications to prevent an eavesdropper from distinguishing an ongoing transmission from noise.

We use the CJ as the source of adversarial perturbation to manipulate the classifier at an eavesdropper into making classification errors. While a perturbation with high power level transmitted by the CJ can easily fool the classifier, it would also increase the interference and the bit error rate (BER) at the intended receiver to an unacceptable level. Therefore, an upper bound on the perturbation strength is imposed. A special case of our setting has been considered in [34], where the transmitter with a single antenna adds perturbations to its own signals to fool an eavesdropper with a modulation classifier while aiming to maintain its own communication performance. In this paper, our focus is on covert communications aided by a CJ, whose position can further boost the impact on the eavesdropper to classify received signal as noise while reducing the impact on the BER performance. Note that we only consider fooling a classifier into misclassifying a signal as noise since it is typically more demanding. Further, we extend the analysis to the use of *multiple antennas* at the CJ to generate multiple concurrent perturbations over different channel effects (subject to a total power budget) for better covert communications. This problem setting is different from computer vision applications of adversarial attacks that are limited to a single perturbation that is directly added to the input of a deep neural network (DNN). We assume that the CJ has multiple antennas to transmit adversarial perturbations against the eavesdropper and aims to decrease the probability of detection at the eavesdropper.

In this paper, we design a white-box attack at the CJ where the signal of the CJ is time-aligned with the transmitted signal and uses the maximum received perturbation power (MRPP) attack that was introduced in [20]. We propose different methods to allocate power among antennas at the CJ and to exploit the channel diversity. We first propose a genie-aided adversarial attack where the CJ selects one antenna to transmit the perturbation such that it would result in the worst classification performance depending on the channel condition over the entire symbol block (that corresponds to the input to the DNN at the receiver). Then, we consider transmitting with all the antennas at the adversary where the power allocation is based on the channel gains, either proportional or inversely proportional to the channel gains. Finally, we propose the elementwise maximum channel gain (EMCG) attack to utilize the channel diversity more efficiently by selecting the antenna with the best channel gain at the symbol level to transmit perturbations.

For the performance evaluation, we first consider a CJ with a single antenna using basic modulated signals (e.g., QPSK and 16-QAM), and then extend the setting to a more complicated 5G communication signal. Our results show that we can effectively hide these signals from an eavesdropper that uses a DL-based classifier to detect transmissions. Then, we use multiple antennas at the CJ to investigate the performance of multiple concurrent perturbations over different channel effects on the eavesdropper’s classifier.

During simulations, the perturbation of the CJ is selected to minimize the strength of the perturbation subject to the condition of successfully fooling the eavesdropper and an upper bound on the perturbation power that can translate to limiting the BER at the receiver. We show that Gaussian noise is not effective as an adversarial perturbation and develop an algorithm to optimize the perturbations for the CJ to enable covert communications, which we demonstrate for signals with different modulation types and 5G communications. Furthermore, we show that the EMCG attack outperforms other attacks and effectively uses the channel diversity provided by multiple antennas to cause misclassification at the receiver. This attack improvement remains effective regardless of the channel variance or correlation between channels, whereas the proportional to the channel gain (PCG) attack is greatly affected by the correlation between channels. Finally, we show that increasing the number of antennas at the adversary significantly improves the attack performance by better exploiting the channel diversity to craft and transmit adversarial perturbations.

In summary, our contributions are given as follows:We present how a CJ is used to make wireless communications covert by transmitting adversarial attack against the classifier of the eavesdropper.For a CJ equipped with multiple antennas, we investigate the use of multiple antennas to generate multiple concurrent perturbations over different channel effects against the eavesdropper. Furthermore, we propose different methods to utilize the channel diversity.With simulations, we show that the CJ can generate perturbation signals that cause misclassification at the eavesdropper for both basic modulated signals and sophisticated 5G signals, while the BER at the receiver is slightly affected.

The rest of the paper is organized as follows. Section 2 describes the system model. Section 3 presents the white-box adversarial attacks when the CJ has one antenna. Section 4 introduces different methods to generate adversarial attacks when the CJ has multiple antennas. Section 5 presents the performance evaluation results. Section 6 concludes the paper.

## 2. System Model

We consider a wireless system that consists of a transmitter, a receiver, a CJ, and an eavesdropper as shown in Figure 1. The transmitter sends *p* complex symbols consecutively in time, $x\in C^{p}$, by mapping a binary input sequence $m\in {\left\{0,1\right\}}^{l}$. Specifically, $x=g_{s}\left(m\right)$, where $g_{s}:{\left\{0,1\right\}}^{l}\to C^{p}$ and *s* represents the modulation type of the transmitter. Then, the transmitter’s signal received at node *j* (either the receiver *r* or the eavesdropper *e*) is given by
(1)$$\begin{array}{c} r_{tj}=H_{tj}g_{s}\left(m\right)+n_{tj}=H_{tj}x+n_{tj},\;j\in \left\{r,e\right\}, \end{array}$$
where $H_{tj}=diag\left\{h_{tj,1},\cdots ,h_{tj,p}\right\}\in C^{p\times p}$ and $n_{tj}\in C^{p}$ are the channel and complex Gaussian noise from the transmitter to node *j*, respectively. Upon receiving the signal $r_{tr}$, the receiver decodes the message with the BER given by
(2)$$P_{e}\left(m,r_{tr}\right)=\frac{1}{l}\sum_{i=1}^{l}I\left\{m_{i}\neq {\hat{m}}_{i}\right\},$$
where ${\hat{m}}_{i}$ is a decoded bit and I{·} is an indicator function.

The eavesdropper tries to detect the existence of wireless transmission using a pre-trained DL-based classifier, namely a DNN, $f\left(.,\theta \right):X\to R^{2}$, where θ is the set of DNN parameters and $X\subset C^{p}$. An input x∈X is assigned a label $\hat{l}\left(x,\theta \right)=\arg\max_{k}f_{k}\left(x,\theta \right)$, where $f_{k}\left(x,\theta \right)$ is the output of a classifier *f* corresponding to the *k*th class.

To make communications between the transmitter and its receiver covert, the CJ with *q* antennas transmits perturbation signals $\delta_{1},\delta_{2},\cdots ,\delta_{q}\in C^{p}$, where the *i*th antenna transmits $\delta_{i}$, to cause misclassification at the eavesdropper by changing the label of the received signal $r_{te}$ from *signal* to *noise*. Thus, if the transmitter transmits x, the received signal at node *j* is given by
(3)$$\begin{array}{cc} \; & r_{tj}^{\prime }\left(\delta_{1},\cdots ,\delta_{q}\right)=H_{tj}x+\sum_{i=1}^{q}H_{c_{i}j}\delta_{i}+n_{tj},\;j\in \left\{r,e\right\}, \end{array}$$
where $H_{c_{i}j}=diag\left\{h_{c_{i}j,1},\cdot \cdot \cdot ,h_{c_{i}j,p}\right\}\in C^{p\times p}$ is the channel from the *i*th antenna of the CJ to node *j*.

Since the perturbation signals from the CJ not only creates interference at the eavesdropper, but also at the receiver, the CJ determines its signals $\delta_{1},\delta_{2},\cdots ,\delta_{q}$ to cause misclassification at the eavesdropper using a fixed power budget $P_{max}$ that also limits the BER at the receiver. Formally, the CJ first determines $\delta_{1},\delta_{2},\cdots ,\delta_{q}$ by solving the following optimization problem:(4)$$\begin{array}{cc} \arg\min_{\delta_{i}} & \;\sum_{i=1}^{q}\left|\right|\delta_{i}{\left|\right|}_{2}^{2}\\ s.t. & \;\hat{l}\left(r_{te},\theta \right)\neq \hat{l}\left(r_{te}^{\prime }\left(\delta_{1},\cdots ,\delta_{q}\right),\theta \right)\\ & \;\sum_{i=1}^{q}\left|\right|\delta_{i}{\left|\right|}_{2}^{2}\leq P_{max}. \end{array}$$

The solution $\delta_{i}^{*}$ to (Equation 4) results in a BER, $P_{e}\left(m,r_{tr}^{\prime }\left(\delta_{i}^{*}\right)\right)$, at the receiver that can be bounded to a target level by selecting $P_{max}$ accordingly. Since solving (Equation 4) is difficult, different methods have been proposed in computer vision to approximate the adversarial perturbations such as the fast gradient method (FGM) [7]. The FGM is computationally efficient for crafting adversarial attacks by linearizing the loss function, L(θ,x,y), of the DNN classifier in a neighborhood of x where y is the label vector. This linearized function is used for optimization. In this paper, we consider a *targeted attack*, where the perturbation of the CJ aims to decrease the loss function of the label *noise* and cause a specific misclassification, from *signal* to *noise*, at the eavesdropper even though there is an actual transmission. We approach the problem from an AML point of view and aim to fool a target classifier, which is equivalent to hiding communications in noise from a wireless communications perspective. While designing the perturbation, we constrain the BER at the receiver to stay below a certain level while satisfying the power constraint at the CJ, as stated in the constraints of the optimization problem (Equation 4). We assume that the CJ collaborates with the transmitter and thus knows the transmitted signal from the transmitter.

## 3. Adversarial Perturbation for the CJ

In this section, we design the white-box perturbation for the CJ using a targeted FGM to solve (Equation 4). We first assume that the CJ has one antenna, q=1. We will relax the assumption in Section 4. For the targeted attack, the CJ minimizes $L(\theta ,r_{te}^{\prime }\left(\delta \right),y^{target})$ with respect to δ where $y^{target}$ is the one-hot-encoded desired target class. We fix $y^{target}$ as *noise* label since the CJ always tries to add perturbation to fool the eavesdropper into misclassifying a received signal as noise. We use FGM to linearize the loss function as $L\left(\theta ,r_{te}^{\prime }\left(\delta \right),y^{target}\right)\approx L\left(\theta ,r_{te},y^{target}\right)+{\left(H_{ce}\delta \right)}^{T}\nabla_{x}L\left(\theta ,r_{te},y^{target}\right)$ and then minimize it by setting $H_{ce}\delta =-\alpha \nabla_{x}L\left(\theta ,r_{te},y^{target}\right)$, where α is a scaling factor to constrain the adversarial perturbation power to $P_{max}$. The details of determining the CJ’s perturbation signal are presented in Algorithm 1. After we obtain the δ that causes misclassification at the eavesdropper and satisfies the power constraint, we check the BER at the receiver. The perturbation power can further be adjusted to meet a target BER level. Specifically, if the BER level at the receiver is more important than fooling the eavesdropper, we can decrease the adversarial perturbation power. On the other hand, if fooling the eavesdropper is the priority, we can increase the adversarial perturbation power.  
**Algorithm 1:** Generating the perturbation of the CJ
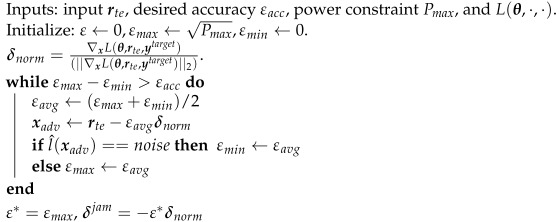


## 4. Adversarial Perturbations Using Multiple Antennas at the CJ

In this section, we present different methods to utilize *q* antennas at the CJ to improve the performance of the adversarial attack against the eavesdropper. Note that the adversary can allocate power differently to each antenna and increase the channel diversity by using multiple antennas. In this paper, we apply the targeted MRPP attack in [20], which has been developed from the attack in [15] by accounting for additional channel effects.

### 4.1. Single-Antenna Genie-Aided (SAGA) Attack

We first begin with an attack where the CJ allocates all the power to only one antenna for the entire symbol block of an input to the classifier at the eavesdropper as shown in Figure 2a. In this attack, we assume that the CJ is aided by a genie and thus knows in advance the best antenna out of *q* antennas that causes a misclassification. Then, the genie-aided CJ puts all the power to that one specific antenna to transmit the adversarial perturbation against the eavesdropper.

### 4.2. Proportional to Channel Gain (PCG) Attack

To exploit the channel with the better channel gain, the CJ allocates more power to better channels. Specifically, the power allocation for the *i*th antenna is proportional to the channel gain $∥h_{c_{i}e}{∥}_{2}$, where $h_{c_{i}e}={\left[h_{c_{i}e,1},\cdot \cdot \cdot ,h_{c_{i}e,p}\right]}^{T}$, using weight $w_{i}=\frac{∥h_{c_{i}e}{∥}_{2}}{\sum_{j=1}^{q}{∥h_{c_{j}e}∥}_{2}},i=1,\cdots ,q$. The adversarial perturbation that is transmitted by each antenna is generated using the MRPP attack as before and transmitted with the power allocated to each antenna. The detailed algorithm is presented in Algorithm 2.   
**Algorithm 2:** PCG attack
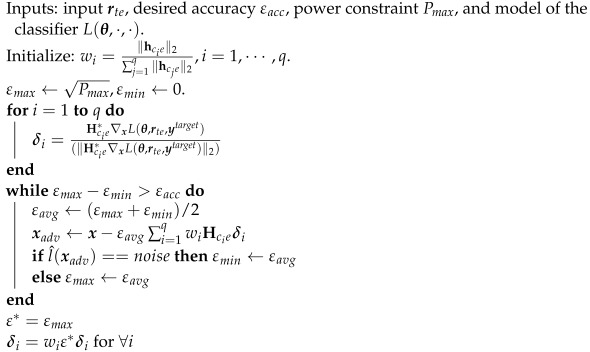


### 4.3. Inversely Proportional to Channel Gain (IPCG) Attack

In contrast to the PCG attack, the CJ allocates more power to weak channels to compensate for the loss over the weak channels, i.e., inversely proportional to the channel gain. The perturbations that are transmitted by each antenna are generated using the MRPP attack and the power for each antenna is determined to be inversely proportional to the channel gain. The algorithm is the same as Algorithm 2 except that $w_{i}$ changes to be inversely proportional to the channel, i.e., $w_{i}=\frac{1}{∥h_{c_{i}e}{∥}_{2}}\frac{1}{\sum_{j=1}^{q}\frac{1}{∥h_{c_{j}e}{∥}_{2}}},i=1,\cdots ,q$.

### 4.4. Elementwise Maximum Channel Gain (EMCG) Attack

Unlike the previous attacks that considered the channel gain of the channel vector with dimension p×1 as a way to allocate power among antennas, the EMCG attack considers the channel gain for each time instance to fully utilize the channel diversity as shown in Figure 2b. First, the CJ compares the channel gains elementwise and selects one antenna that has the largest channel gain at each instance. Specifically, the CJ finds and transmits with the antenna $j^{*}=\arg\max_{j=1,\cdots ,q}\{∥h_{ar_{j},t}{∥}_{2}\}$ that has the largest channel gain at instance *t*. Furthermore, a virtual channel $h_{vir,t}$ at instance *t* is defined as the channel with the largest channel gain among antennas which is $h_{ar_{j^{*}},t}$. Then, the adversary generates the perturbation $\delta^{vir}$ with respect to $h_{vir}={\left[h_{vir,1},\cdots ,h_{vir,p}\right]}^{T}$ using the MRPP attack and transmits each element of $\delta^{vir}$ with the antenna that has been selected previously. The details are provided in Algorithm 3.
**Algorithm 3:** EMCG attack
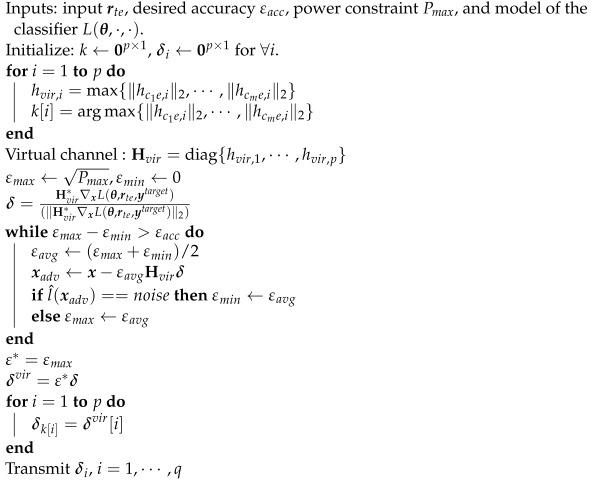


## 5. Simulation Results

We analyzed the success of covertness achieved by CJ’s perturbation at the eavesdropper and the corresponding effect on the BER at the receiver. We first assumed that the CJ only had one antenna to analyze the impact of the CJ on the eavesdropper. Then, we increased the number of antennas at the CJ to observe the performance when multiple antennas are used with different methods. We compared this perturbation with random Gaussian noise transmitted by the CJ. Furthermore, we changed the location of the CJ to investigate the effects of topology and channel.

### 5.1. Simulation Settings

We assumed that the binary source data were generated independently and uniformly at the receiver. The classifier at the eavesdropper was a convolutional neural network (CNN). The input to the CNN was of two dimensions (2,16) corresponding to 16 in-phase/quadrature (I/Q) data samples. The CNN consisted of a convolutional layer with kernel size (1,3), a hidden layer with dropout rate 0.1, a rectified linear unit (ReLU) activation function at the convolutional and hidden layers and a softmax activation function at the output layer that provides the label *signal* or *noise*. We applied a backpropagation algorithm with the Adam optimizer to train the CNN using cross-entropy as the loss function. The CNN was implemented in Keras with the TensorFlow backend. We assumed that the eavesdropper already knew the signal type that was used at the transmitter. Thus, the classifier at the eavesdropper was only trained with two labels, *signal* and *noise*. For each signal type, we trained a separate classifier using different datasets, where 20,000 symbols were generated and split into blocks of 16 I/Q symbols. The channel between the nodes had path-loss effects and Rayleigh fading such that the channel gain from node *i* to node *j* was $h_{ij}={\left(\frac{d_{0}}{d_{ij}}\right)}^{\gamma }h_{i,j}$, where $d_{ij}$ is the distance from node *i* to *j*, $d_{0}$ is the reference distance, $h_{i,j}$ is Rayleigh fading between node *i* to *j*, and γ is the path loss exponent. We set $d_{0}=1$ and γ=2.8 throughout the simulations. Note that there was only a path loss component in the channels for the simulations with CJ for the case of one antenna.

We used the perturbation-to-noise ratio (PNR) metric from [15] that captures the relative perturbation power at the CJ with respect to the noise and measured how the increase in the PNR affected the accuracy of the classifier at the eavesdropper. As the PNR increases, the perturbation generated by the CJ is more likely to be detected by the eavesdropper and increases the BER at the receiver.

### 5.2. Performance Evaluation of CJ with One Antenna for Signals with Different Modulations

We first assumed that the CJ only had one antenna, q=1, and aimed to hide signals with a fixed modulation scheme, namely QPSK or 16QAM, used by the transmitter using Algorithm 1. Note that we used only Algorithm 1 since the CJ only had a single antenna. The first topology that we considered was $d_{cr}=d_{ce}=1$. In Figure 3, we show how the perturbation signal generated by the CJ affects the classifier at the eavesdropper. The *x*-axis is the PNR (measured in dB) and the *y*-axis is the success of covertness (measured in percentage) that indicates the success of making wireless communications covert, namely the likelihood that the eavesdropper classifies a signal plus perturbation as noise. We observe that as the SNR of the signal increases, the CJ needs more perturbation power to cause misclassification at the eavesdropper. Furthermore, the 16QAM-modulated signal is more susceptible to adversarial perturbation than the QPSK-modulated signal, since it is more difficult to distinguish the 16QAM-modulated signal from the noise for the same SNR. Furthermore, we observe that the success of covertness suddenly increases after some PNR value for both modulation types. On the contrary, the Gaussian noise based perturbation has negligible effect on the classifier for all SNR values. We further observe that the Gaussian noise with more power decreases the success of covertness when the SNR of 16-QAM modulated signal is 3 dB. The reason is the Gaussian noise strengthens the noise which makes the received signal at the eavesdropper resemble the strength of the signal, thus the classifier at the eavesdropper classifies the received signal as signal.

In Figure 4, we consider $d_{cr},=1.5$ and $d_{ce}=0.5$ (namely, the distance between the CJ and the receiver is increased and the distance between the CJ and the eavesdropper is decreased compared to Figure 3). As the SNR of the signal increases, the CJ requires more power to cause misclassification at the eavesdropper, as we also observed in Figure 3. Due to the reduced path loss effect between the CJ and the eavesdropper, less power is required to cause misclassification compared to Figure 3. This result motivates the use of AML instead of conventional jamming (e.g., [51]) to attack an eavesdropper.

#### Reliability of Communications

The BER performance at the receiver for different modulation types and SNR values is compared in Figure 5 when $d_{cr}=d_{ce}=1$. We observe that the BER of the 16QAM-modulated signals is more susceptible to the adversarial perturbation signal than the BER of QPSK-modulated signals. The reason is that since the 16QAM transmits more bits than the QPSK per symbol, the distances between constellation points are smaller, which leads to a larger BER for a given SNR. Moreover, as the SNR increases, the average BER decreases as expected. For the CJ with the proposed adversarial perturbation, we observe that the BER curve saturates after some PNR value because the successful perturbation signal can be generated using less power than the maximum power that the CJ can use. Figure 5 can be used as a guideline to determine the maximum PNR to satisfy the BER requirement at the receiver. For example, to meet the target BER of 0.15 for a QPSK-modulated signal, the PNR is selected to be at most −8 dB when the SNR is 3 dB and the resulting success of covertness is 65%. Furthermore, we observe that the Gaussian noise based perturbation results in a lower BER than the adversarial perturbation in the low PNR regime. However, the BER gap between these two CJ schemes decreases when the PNR increases, and the adversarial perturbation results in a smaller BER in the high PNR region.

The BER performance at the receiver for different modulation types and SNR values is compared in Figure 6 when $d_{cr}=1.5$ and $d_{ce}=0.5$. We observe that the BER gap between the Gaussian noise and adversarial perturbation for the same SNR value decreases due to the increased path loss effect between the CJ and the receiver. Thus, the CJ can create a perturbation signal that causes misclassification with higher success without increasing the BER further if the location of the CJ is closer to the eavesdropper. This result motivates the control of the CJ positions to fool a target classifier while protecting the BER performance of the intended receiver.

### 5.3. Performance Evaluation for 5G Communications

As a full-fledged waveform to hide, we considered the 5G physical layer communications where a 5G user equipment (UE) transmits a 5G uplink signal to a base station (gNodeB) in the presence of the perturbation from the CJ. MATLAB’s 5G toolbox was used to generate 5G signals that included the transport (uplink shared channel, UL-SCH) and physical channel. The transport block was segmented after the cyclic redundancy check (CRC) addition and low-density parity-check (LDPC) coding was used as forward error correction. The output codewords were QPSK-modulated as an example. Next, the generated resource grid was OFDM-modulated with inverse fast Fourier transform and cyclic prefix (CP) addition operations where the subcarrier spacing was 15 kHz. The target code rate was set to $\frac{820}{1024}$ and the output I/Q samples were stored after the signal passed through the channel. The eavesdropper attempted to distinguish the received signals from noise, whereas the receiver attempted to decode the received signals by removing the CP and performing FFT, channel equalization, QPSK demodulation, LDPC, and CRC decoding operations.

#### 5.3.1. Covertness of Communications

The success of covertness for 5G communications is considered in Figure 7. As in the previous figures for QPSK-modulated signals and 16QAM-modulated signals, the proposed perturbation outperforms the Gaussian noise significantly in the high-PNR region for 5G signals. Furthermore, we observe that more power is needed for the CJ to fool the classifier at the eavesdropper when the distance between the CJ and the eavesdropper increases.

#### 5.3.2. Reliability of Communications

The BER for 5G communications is shown in Figure 8. When $d_{ce}=d_{cr}=1$ and the SNR is 5 dB, the Gaussian noise based perturbation has a higher BER performance compared to the proposed perturbation and a similar result is also observed for other SNR values. Note that the adversarial perturbation by the CJ not only increases the success of covertness, but also has less effect on the BER performance of the receiver compared to the Gaussian noise based perturbation for 5G communication signals. We further observe that the Gaussian noise based perturbation results in a higher BER than the proposed adversarial perturbation when $d_{ce}=0.5$ and $d_{cr}=1.5$.

### 5.4. Performance Evaluation of CJ with Multiple Antennas

Next, we analyzed the performance of the CJ with multiple antennas when a QPSK-modulated signal was used at the transmitter. Note that the channel between the CJ and the receiver and the channel between the CJ and the eavesdropper had Rayleigh fading. Note that $h_{i,j}∼\mathrm{Rayleigh}\left(0,1\right)$ if specified otherwise. Figure 9a presents the success of covertness when the CJ transmits an adversarial perturbation with q=2 antennas using the different attack methods introduced in Section 4. We observe that all different methods using multiple antennas outperform the attack generated by the CJ with one antenna. Furthermore, randomly selecting one antenna at the CJ performs worst among attacks using multiple antennas and the performance of the IPCG attack is similar to the performance of the PCG attack. Moreover, the EMCG attack outperforms other attacks by fully utilizing the channel diversity.

Figure 9b presents the BER performance of different attack methods. We observe that the CJ using one antenna gives the largest BER whereas the PCG and IPCG attacks give the smallest BER. Furthermore, the EMCG attack gives a moderate BER increase while successfully making communications covert.

The performance of the CJ with different number of antennas is presented in Figure 10a. As the number of the antennas at the CJ increases, the success of covertness also increases suggesting that using more antenna at the CJ helps the covertness of communications. Furthermore, the BER decreases when more antennas are used at the CJ as we can see from Figure 10b. Therefore, using more antennas at the CJ is always beneficial for communications in terms of covertness and BER when the EMCG attack is used at the CJ.

Next, we varied the SNR levels to analyze how the SNR affected the covertness and the BER in Figure 11a,b. As expected, the CJ needs a higher PNR to fool the eavesdropper when the SNR is high. Furthermore, we observe that the BER slightly increases when the PNR increases and the BER is higher for a lower SNR.

Finally, we increased the variance of the Rayleigh fading between the CJ and the eavesdropper to analyze the effect of the channel on the covertness of communications. In Figure 12a, we observe that a lower PNR is needed to fool the eavesdropper when the variance of the Rayleigh fading is high. Furthermore, as a consequence of using a lower PNR at the CJ, a higher variance of Rayleigh fading results in a lower BER at the receiver.

## 6. Conclusions

We considered a wireless communications system in which a CJ with multiple antennas transmits perturbation signals to fool a DL-based classifier at the eavesdropper into classifying the ongoing transmissions as noise. Following the AML approach, the CJ was designed to generate the perturbation signal with different methods. For both basic modulated signals and sophisticated 5G signals, we showed that the CJ could generate a perturbation signal that caused misclassification at the eavesdropper (from *signal* to *noise*) with high success, while the BER at the receiver was only slightly affected. Furthermore, we showed that by adding more antennas at the CJ always improved the attack performance and lowered the BER when the EMCG attack was used. These results demonstrate that wireless communications can be successfully kept covert when multiple antennas are used at the CJ by allocating the transmit power efficiently.

## Figures and Tables

**Figure 1 entropy-24-01047-f001:**
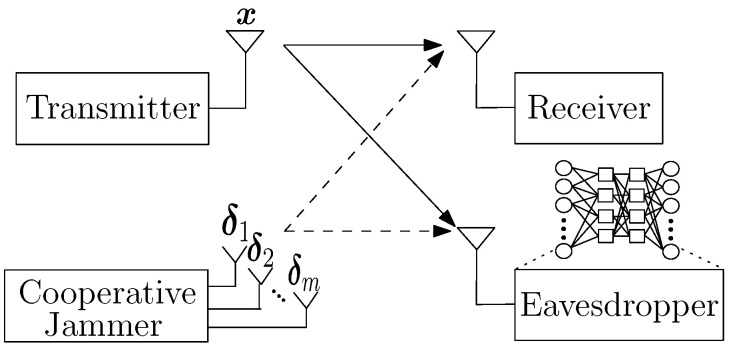
System model.

**Figure 2 entropy-24-01047-f002:**
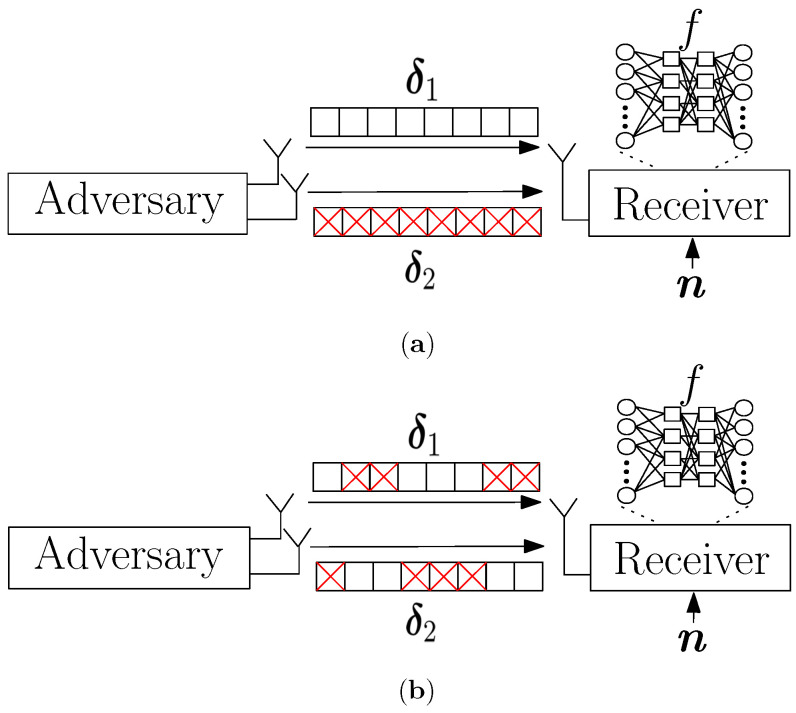
Illustration of (**a**) SAGA attack and (**b**) EMCG attack.

**Figure 3 entropy-24-01047-f003:**
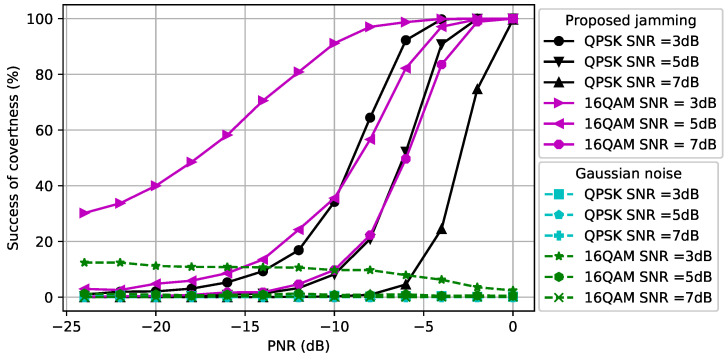
Success of covertness at the eavesdropper when $d_{ce}=d_{cr}=1$.

**Figure 4 entropy-24-01047-f004:**
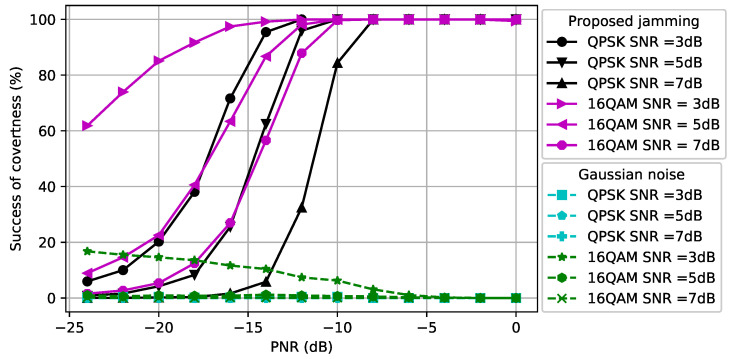
Success of covertness at the eavesdropper when $d_{ce}=0.5$ and $d_{cr}=1.5$.

**Figure 5 entropy-24-01047-f005:**
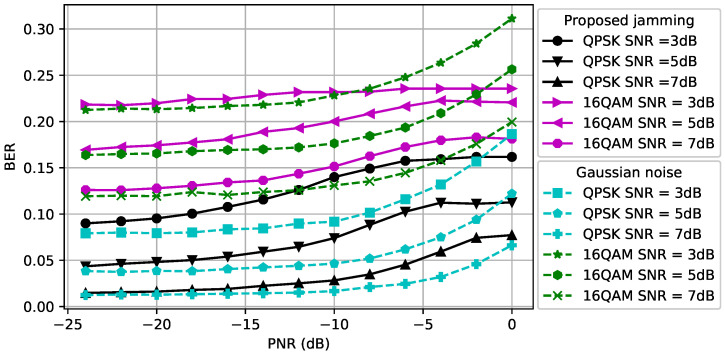
BER at the receiver when $d_{ce}=d_{cr}=1$.

**Figure 6 entropy-24-01047-f006:**
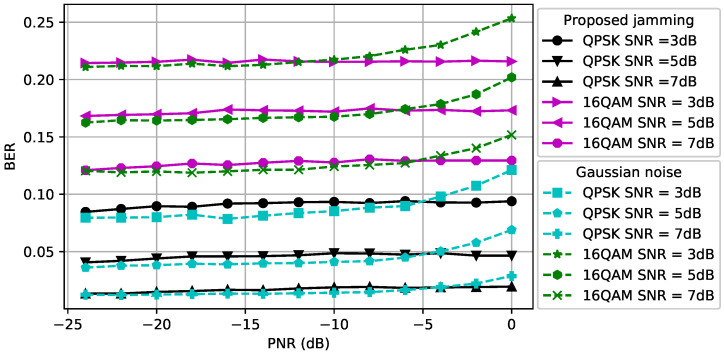
BER at the receiver when $d_{ce}=0.5$ and $d_{cr}=1.5$.

**Figure 7 entropy-24-01047-f007:**
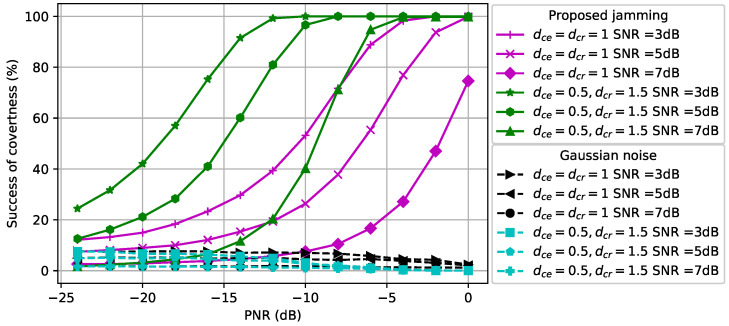
5G communications covertness performance at the eavesdropper.

**Figure 8 entropy-24-01047-f008:**
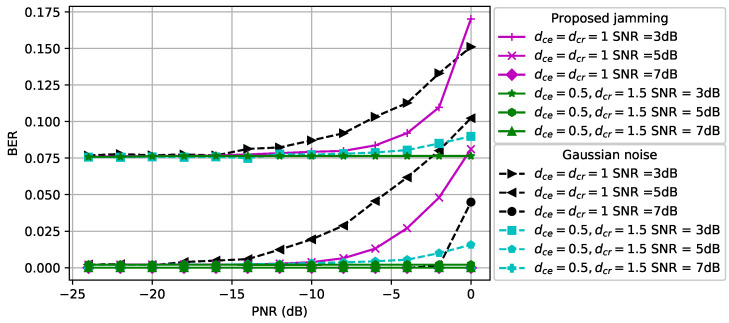
5G communications BER performance at the receiver.

**Figure 9 entropy-24-01047-f009:**
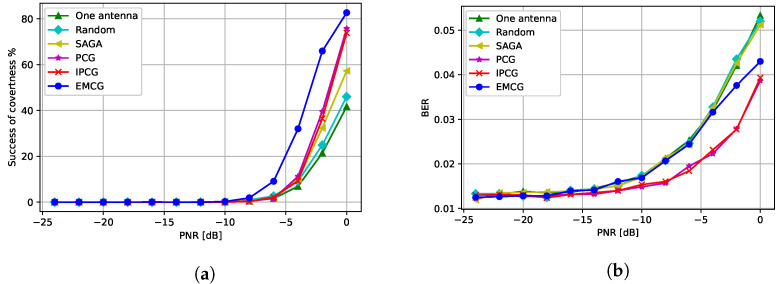
Performance when CJ has q=2 antennas: (**a**) success of covertness and (**b**) BER at the receiver.

**Figure 10 entropy-24-01047-f010:**
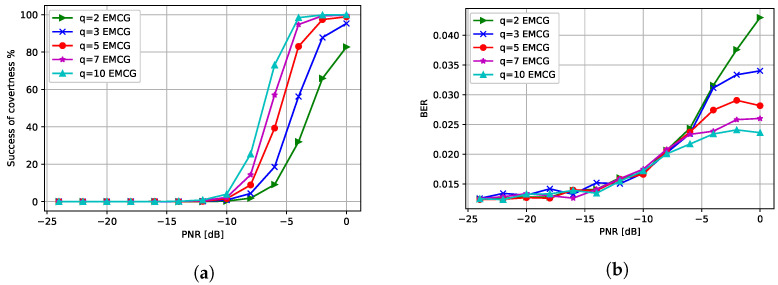
Performance with different number of antennas at the CJ: (**a**) success of covertness and (**b**) BER at the receiver.

**Figure 11 entropy-24-01047-f011:**
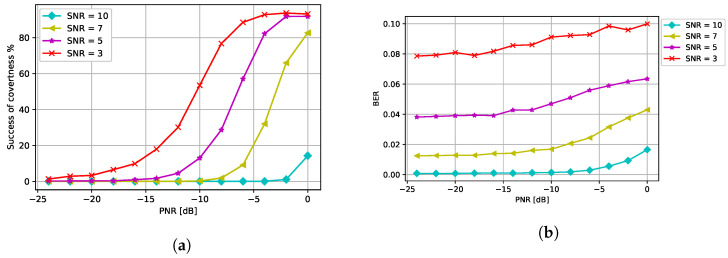
Performance with respect to different SNR levels: (**a**) success of covertness and (**b**) BER at the receiver.

**Figure 12 entropy-24-01047-f012:**
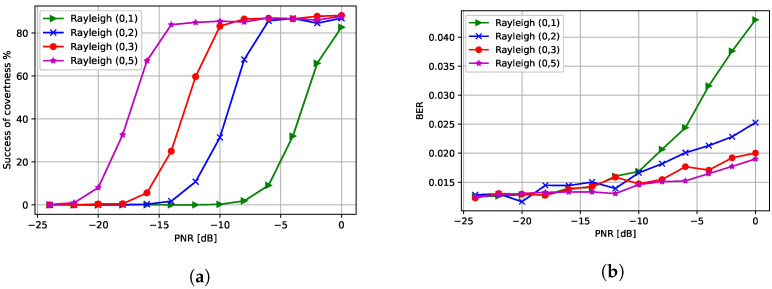
Performance with respect to different Rayleigh fading variances: (**a**) success of covertness and (**b**) BER at the receiver.

## Data Availability

Not applicable.

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
