# Peer review of "Adversarial Machine Learning for NextG Covert Communications Using Multiple Antennas"

_entropy, 2022, doi:10.3390/e24081047_

Round 1

Reviewer 2 Report

The authors proposed a novel adversarial signal design for covert communication to show that careful design that accounts for MIMO channel can significantly improve the system security. The idea of fully employing the best channels to achieve the strongest received adversarial signal given the power constraint was interesting to the reviewer. Please find some of my minor comments below.

1. Regarding Eq 1), for the x being in C^p, it would be nice to mention that p is the dimension of some orthogonal resources, e.g., time or frequency. For the readers who are not familiar with this topic may think that x has p transmit antennas.

2. In line 125, there is a typo: f(\cdot, \theta) since ';' is not used afterwards.

3. In eq 4), it would be nice to write \argmin_{\delta_1,\cdots,\delta_q}.

4. For the Fig. 3 ~8, is would be nice to detail the legend for the proposed scheme, e.g., 'Proposed jamming (EMCG)'. 

5. Lastly, it would be nice to give some intuition as a future work that, what would be the nice strategy for adversarial signal design when transmit and receive antennas are both equipped with multi-antennas.

Overall, the paper is clearly well-written with strong experimental verifications.

Thank you.

Round 2

Reviewer 1 Report

The reviewer thanks the authors for their efforts on editing the manuscript.